# Trait-Based Vaccination of Individual Meerkats (*Suricata suricatta*) against Tuberculosis Provides Evidence to Support Targeted Disease Control

**DOI:** 10.3390/ani12020192

**Published:** 2022-01-13

**Authors:** Stuart J. Patterson, Tim H. Clutton-Brock, Dirk U. Pfeiffer, Julian A. Drewe

**Affiliations:** 1Veterinary Epidemiology, Economics and Public Health Group, Royal Veterinary College, University of London, Hawkshead Lane, Hatfield, Hertfordshire AL9 7TA, UK; dirk.pfeiffer@cityu.edu.hk (D.U.P.); jdrewe@rvc.ac.uk (J.A.D.); 2Large Animal Research Group, Department of Zoology, University of Cambridge, Downing Street, Cambridge CB2 3EJ, UK; thcb@cam.ac.uk; 3Mammal Research Institute, University of Pretoria, Hatfield, Pretoria 0028, South Africa; 4Department of Infectious Diseases and Public Health, Jockey Club College of Veterinary Medicine and Life Sciences, City University of Hong Kong, Hong Kong, China

**Keywords:** targeted disease control, meerkats, trait-based vaccination, wildlife disease

## Abstract

**Simple Summary:**

There is evidence to show that, within a population, some individuals are more likely to spread infections than others. When trying to protect a population against infection, most strategies aim to vaccinate as many individuals as possible. However, vaccinating wildlife is difficult because individuals are difficult to find and capture. For wildlife therefore, the ideal strategy would involve targeting vaccinations at those individuals most likely to transmit infection, thus gaining maximum benefit from capturing a small number of individuals. Whilst this seems a very attractive solution, very few studies have attempted to provide evidence to support this theory. This study focuses on a population of meerkats with a history of tuberculosis. Previous work has suggested that socially dominant individuals are most likely to transmit infection, with subordinates most likely to become infected. Therefore, whilst some social groups were left untreated as a baseline, in others, either dominants or subordinates were vaccinated. All groups were monitored for two years, after which time the infection data was analysed. Groups in which vaccinations had been used showed reduced infection rates suggesting that the targeted approach had reduced transmission. A targeted approach may therefore offer an efficient option for vaccinating wildlife in the future.

**Abstract:**

Individuals vary in their potential to acquire and transmit infections, but this fact is currently underexploited in disease control strategies. We trialled a trait-based vaccination strategy to reduce tuberculosis in free-living meerkats by targeting high-contact meerkats (socially dominant individuals) in one study arm, and high-susceptibility individuals (young subordinates) in a second arm. We monitored infection within vaccinated groups over two years comparing the results with untreated control groups. Being a member of a high-contact group had a protective effect on individuals’ survival times (Hazard Ratio = 0.5, 95% Confidence Interval, CI: 0.29–0.88, *p* = 0.02) compared to control groups. Over the study, odds of testing positive for tuberculosis increased more than five-fold in control groups (Odds Ratio = 5.40, 95% CI = 0.94–30.98, *p* = 0.058); however, no increases were observed in either of the treatment arms. Targeted disease control approaches, such as the one described in this study, allow for reduced numbers of interventions. Here, trait-based vaccination was associated with reduced infection rates and thus has the potential to offer more efficient alternatives to traditional mass-vaccination policies. Such improvements in efficiency warrant further study and could make infectious disease control more practically achievable in both animal (particularly wildlife) and human populations.

## 1. Introduction

Individuals vary in their ability to transmit and acquire infections. Exploiting such heterogeneities may make disease control more efficient [1]. Most vaccination programmes, however, assume that interactions (and disease transmission) within populations are random and homogenous [2]. This study exploits heterogeneities in social behaviour to investigate whether intra-group transmission of a naturally-occurring pathogen in free-living meerkats (*Suricata suricatta*) can be reduced by the application of a vaccination protocol targeting individual animal traits.

A cornerstone of infection control is the basic reproduction number, R_0_, the average number of secondary cases caused by each primary case in a naive population [3,4]. This value determines the proportion of a population that must be immune in order to reduce disease transmission (Herd Immunity Threshold, HIT), in both human and veterinary medicine [5,6]. When such a transmission reduction is achieved, the remaining susceptible individuals within the population experience an indirect benefit of others’ immunity; they experience less exposure to infection without having being immunised themselves [7]. Whilst the HIT is useful in establishing evidence-based protocols [8], its weakness is that it assumes a homogenous population. Individual variations in social connectivity can lead to small numbers of hosts that are much more likely to transmit infection than others: the 20/80 rule states that 20% of individuals (“superspreaders”) are responsible for 80% of transmissions within a population [1,9]. A vaccination strategy based upon HIT can lead to individuals with high potential for transmitting infection remaining unvaccinated by chance. In 1999, for example, despite 96% of the human population of the Netherlands being vaccinated against measles (the HIT for this disease is estimated to be between 83% and 94% [10]) an outbreak involving around 3000 individuals resulted from just five highly connected initial cases [11]. In theory, if that core of superspreaders could be targeted for infection control, then reductions in population transmission could be achieved with fewer interventions than through blanket HIT methods, thus improving efficiency of control. Empirical trials of such approaches are, however, rare, and especially so in free-living wild animal populations where infectious diseases can pose large threats to domestic animals and humans [6,12,13].

A targeted protocol could reduce the number of individuals needing vaccination, and studies such as that undertaken by Medlock and Galvani [14] in the human field, have focused on the optimal allocation of vaccine doses. This has particular benefits in a wildlife setting [15]. Widespread vaccination of wildlife incurs logistical difficulties [16], such as initial estimation of population numbers, efficiency of delivery, and unintentional repeat doses [17]. Furthermore, live vaccines pose the additional small risk of reverting to virulence, or contributing to the selection of non-vaccinal strains [18]. Reducing the number of individuals requiring vaccination therefore is advantageous; whether such approaches can be effective in the field is, however, unproven and is the focus of the work presented here. Strategies for targeting, may be based upon time and space [19], or upon characteristics of individuals and traits within a population [15].

Targeting individuals based upon social network position has been shown to be effective in theoretical work, though empirical studies in animals are needed. Similarly, studies of such interventions in human populations are predominantly model-based [20]. Rushmore et al. [21] used field data to construct social contact networks of chimpanzees (*Pan troglodytes*) and predicted that the size of an infectious outbreak would vary depending upon the network position of the initial case. It was predicted that a theoretical disease could be controlled with 35% fewer vaccinations when using a trait-based approach than when compared with random vaccination. Implementing such an approach would require a system with a well understood social network structure, heterogeneity within transmission, and knowledge of the infection dynamics within the system. Unfortunately, these factors are unknown in the majority of wild animal populations.

A long-term study of meerkats in South Africa’s Kalahari Desert does, however, provide a sufficiently characterised system. Network studies of both inter- and intra-group contacts have been conducted [22,23,24,25], and investigations into how individual characteristics relate to the likelihood of disease transmission have been performed [25,26]. This population is habituated to close human contact, allowing researchers to make ongoing observations of individually identifiable animals [27]. Meerkats live in social groups of 3–50 individuals [28,29], foraging communally by day and sharing underground sleeping burrows at night [30]. Median life expectancy for an individual has been estimated at 2.3 years [31]. Their social hierarchy leads to variety in the behaviours of individuals, with animals either being dominant within a group, or occupying a subordinate position [27,32]. Both aggression and grooming behaviours have been demonstrated to be associated with tuberculosis (TB, caused by *Mycobacterium suricattae*, a novel strain of the *Mycobacterium tuberculosis* complex [33]) using social network analysis [25]. This disease is endemic in this population, causing individual fatalities, after a latent period estimated at around 12 months [24], and group extinctions continuously since 2001 [26]. Those individuals that spend a greater amount of time grooming, and that receive a higher level of aggression (subordinate individuals in both cases), have been shown to be at greater risk of acquiring infection than those dominant individuals who are the recipients of grooming and who reinforce their social position through aggressive behaviours [25]. The characteristics of this study population makes it an excellent candidate for an empirical field trial of a targeted control intervention based upon individual animal traits.

In this cohort study, a field intervention trial was conducted using two trait-based vaccination strategies within the Kalahari meerkat population. In order to ascertain whether such strategies could successfully reduce disease transmission, we asked whether these interventions were associated with any reduction in TB transmission within social groups. In the high contact (HC) arm of the study, the individuals with typically the highest rates of social contact (dominant individuals in meerkat society) were vaccinated to investigate whether there was an indirect benefit to their groups as a whole. In a separate arm of the study (high-susceptibility, HS), using different social groups, individuals who by their social status have previously been shown to be the most susceptible to disease (the subordinate animals) were targeted. In both instances, survival time and the time until first detected infection for members of these treatment sets were compared to control groups which received no intervention. The purpose of this study was not to compare the two interventions with each other, nor to compare them to a random HIT strategy, but to determine whether there was any empirical evidence to demonstrate that a trait-based system could be effective in a wild population.

## 2. Materials and Methods

### 2.1. Study Design

This field study was carried out at the Kuruman River reserve (26°58′ S, 21°49′ E) in South Africa’s Northern Cape. All animals included in the study were part of the Kalahari Meerkat Project’s (KMP) long-term study of a free-ranging meerkat population, and as such, all were individually identifiable from pre-existing dye marks [34]. The present study commenced on 1 September 2014, and ran until 30 September 2016.

Treatment sets were referred to as: (1) high-susceptibility (HS); (2) high-contact (HC), and (3) control. Nine social groups of meerkats were available for inclusion in the study. The groups were ranked in order of the number of individuals and divided into three tiers; large groups (>20 individuals), medium (10–20), and small (<10). Treatment was randomly allocated within each tier such that each treatment set contained a large, a medium, and a small group. A breakdown of the treatment set compositions is given in Appendix A.

### 2.2. Data Collection

Sampling was carried out in five equal time blocks (Block 1: September–November 2014; Block 2: December 2014–February 2015; Block 3: July–October 2015; Block 4: January–March 2016; Block 5: July–September 2016) and sampling of all available individuals within the study was attempted during each block. At each sampling event, an individual was manually caught and sampled under general anaesthesia using isoflurane as described by Drewe et al. [35]. Blood (0.5 mL) was collected by jugular venipuncture for serological analysis using a rapid lateral flow test, DPP VetTB (Chembio Diagnostic Systems, Inc., Medford, New York, NY, USA), and for an Interferon gamma (IFNγ) inducible-protein 10 release assay (IPRA) [36]. A tracheal lavage was also performed to obtain a sample for mycobacterial culture [35]. Pups were first sampled on emergence from the burrow, normally at around 3–4 weeks of age. No anaesthesia was used in pups, and so blood was collected by tail tipping [37] rather than the jugular vein for the safety of the animal. Consequently, tracheal lavage was not possible and a lower volume of blood was available, meaning only the DPP VetTB test was performed on these animals. If an individual was being captured for any other reason (e.g., euthanasia), then opportunistic samples were taken at this point.

All meerkat groups were visited by an observer on at least three days each week. The observer checked all individuals within a group at each visit for visual signs of disease (persistently enlarged submandibular lymph nodes are typical of TB in this species [38]). The presence of individuals within the group was recorded at each visit, and an animal was considered to have died if it was not observed for a period of three months. All births were recorded. Dominance was assigned to a single male and female within each group based upon behavioural observations, and any changes in this status were recorded [37,39,40].

### 2.3. Treatments

From September 2014 onwards, all animals born into the HS treatment groups were vaccinated at the time of their first emergence from the burrow, around 3–4 weeks of age. Dominance is unusual in animals under 2 years, and so these vaccinated pups would make up the subordinate population over the two years of the study. The Bacillus Calmette-Guerin (BCG) vaccine (Pasteur strain 1173P2) was used. In all cases, 1 mL of reconstituted vaccine was injected intramuscularly into the left hind leg. In the HC groups, both the dominant male and female, and the highest-ranking subordinate female, were selected for vaccination; these individuals having been previously shown to have the highest rate of aggressive contacts, a conduit for disease transmission [24]. These vaccinations were carried out at the first sampling event of the study for each of those individuals. No vaccinations were carried out in the control groups, but this treatment set differed in no other way from either the HC or HS sets. All animals were monitored at every visit for injection site swellings, lameness, behavioural changes, and fluctuations in bodyweight.

In the HC groups, on occasions where a vaccinated individual left the study group, and was replaced in the hierarchy by a new animal, the replacement individual was vaccinated at its next sampling event in order to maintain treatment continuity.

### 2.4. Data Analysis

#### 2.4.1. Infection Data

Parallel test interpretation was used based upon previous analysis of the available diagnostic tests [41]. An individual was classified as test positive if it fulfilled one or more of the following criteria: (i) The IPRA result, being the difference in optical density between a peptide-stimulated and a control sample (OD^PCHP-nil^), was greater than 0.038 [36]; (ii) The test line intensity was greater than 5.0 Reflective Light Units (RLU) using DPP serology [42]; or (iii) *M. suricattae* was cultured and confirmed by PCR from a tracheal lavage sample [35]. Within each study block, the odds of an individual testing positive for TB within each treatment set were calculated. While test results were included for samples collected at the time of vaccine administration, results from previously vaccinated animals were excluded from the analysis, as it could not be said with confidence that a positive test was due to infection rather than vaccination. No diagnostic work was carried out in the time period prior to study commencement and so results from Block 1 were the most reliable estimation of initial status. The odds of being test positive at the start of the study (Block 1) were compared with those in the final sampling block using odds ratios calculated in Excel [43].

The incidence of new infection was calculated within each set during time Blocks 2 to 5. An incident case was defined as an individual that was test positive within the block, when (i) it had been previously tested at least once, and had always tested negative; or (ii) this was its first collected sample, having been born within the study period. The latter categorisation was based upon the assumption that there is no vertical transmission and that all pups start life uninfected. Once an individual had been classified as positive, it was exempt from incidence calculations in future time blocks; it contributed to neither the numerator nor the denominator. The first sample collected from any individual born prior to 1 September 2014 was excluded from the analysis of incidence, as it could not be confirmed whether infection had been acquired before or after this date. Incident cases were expressed as a proportion of eligible individuals which were classified as positive for the first time, to give an incidence risk for each block. Incidence risk was compared across sampling Blocks 2–5 for the three treatment sets.

Two Cox regressions allowing for time-varying variables [44] were carried out, the first examining an individual’s survival time until the point of first testing positive, and the second investigating individuals’ time until death. For the purposes of this analysis, death was assumed if an individual was not sighted for a three-month period. All survival analysis was carried out using the Survival package in R [45]. In these regression models, age (0–6 months, 6–12 months, 12–24 months, and older than 24 months following categories used by Sharpe [46]) and treatment groups were used as categorical variables, and all other categories were treated as binary variables.

A Cox regression model for time-varying variables was used in order to allow changing age status and group membership. Data was both left and right censored, with animals being born into the population after the start of the study. For both models, a univariable analysis was carried out initially. First order interactions between risk factor variables were checked for and included where *p* < 0.05. Following this, a multivariable analysis was performed including all terms for which *p* < 0.2 in the univariable, adding terms in a forward stepwise process and testing the significance of variable additions to the model using analysis of deviance for a Cox model [45]. The assumptions of the proportional hazards model were checked by plotting the scaled Schoenfeld residuals, and ascertaining that a zero-gradient line could be drawn within the 95% confidence interval of the LOESS (locally estimated scatterplot smoothing)-smoothed line [47].

For the univariable analysis of time until testing positive, the following explanatory variables were included: treatment set, age, sex, previous group history of TB (a binary variable defined as an observation of clinical disease affecting a member of the same social group, pre-dating the sampling point), social group, and occupying the dominant position within a group. An individual was defined as a case at the sampling point at which it was classified as positive for the first time. Animals born before the start of the study period were excluded if their first sample was test positive, as it could not be determined whether they had become infected before or during the study. Vaccinated individuals were able to contribute to the study until the time point at which they were vaccinated, at which point their data was censored. All individuals entered the study either at their point of birth, or on 1 September 2014 if they were born prior to this date. For the purposes of the regression analysis, individuals remained in the study either until they became a case, or until the time of their last sampling event.

#### 2.4.2. Longevity Data

For analysis of time until death, the influences of direct vaccination, age, sex, dominance status, previous social group history of TB, and treatment set were included. Treatment set and group history of disease were both dependent upon an individual’s social group, and so it was not possible to separate a social group effect. Social group was included as a frailty term in the multivariable model only if these other two group variables were excluded from the model due to negligible effect.

### 2.5. Ethical Approval

Permission to vaccinate was granted by the National Department of Agriculture, Forestry and Fisheries in South Africa, under section 20 of the Animal Disease Act. Ethical permission for the study was given by the University of Pretoria. Permission to carry out research in the region was granted by the Northern Cape Department of Environment and Nature Conservation.

## 3. Results

Throughout the study 14 dominant individuals were vaccinated in HC groups, and 52 pups in HS groups. This included intervening in HC groups to vaccinate a new high-contact animal on three occasions because of departures of previously vaccinated animals. By the end of the study (September 2016) the mean proportion of living group members that had received a vaccination in a HC group was 24.0%, whereas this figure was 81.0% for HS groups. Samples taken from dominant adults, showed that 50% (*n* = 14) were already positive for TB when their vaccine was administered. Post vaccine administration animals were observed to limp for up to 10 min after release, but no further effects were observed.

Between the start and the end of the study period the odds of testing positive increased five-fold in the control set (Odds Ratio, OR = 5.4, 95% Confidence Interval, 95CI: 0.94–30.98, *p* = 0.058), with no evidence of an increase found in either treatment set (*p* = 0.63 for HS and *p* = 1.0 for HC, Table 1). Initially, the odds of being test-positive were greater in the control than in the HS set (OR = 4.58, 95CI: 1.18–17.79, *p* = 0.03), but were similar in the two treatment groups.

When first assessed (Block 2) the incidence risk of infection was similar in all groups, but they diverged by the end of the study (Figure 1). Incidence risk increased within the control groups across the two-year study period from 7.1% (95CI: 0.4–35.8) to 80.0% (95CI: 29.9–99.0) whilst incidence in the HS and HC treatment groups remained lower peaking at 22.2% (95CI: 4.0–59.8) in mid-2015, and 38.1% (95CI: 19.0–61.3) in early 2016, respectively (Figure 1).

Based upon visually observed cases as opposed to diagnosed infection, there were more clinical cases of TB in the meerkat population during the second year of the study (September 2015–August 2016) than the first (September 2014–August 2015), across the entire population, whilst the HC treatment set had the greatest number of cases of the three study arms (Appendix A). A single group in the HS treatment set was excluded from all analyses, because it (Group F) was lost due to a TB outbreak within the first six months of the study. No litters were born in this group, and so it was not possible to administer any treatments, although routine sampling continued. Changes in group sizes and any dominance changes are shown in Appendix A.

Neither vaccination strategy was found to have had an influence on time to infection, based on analysis of 221 animals, of which 55 (24.9%) converted to a test-positive status during the study. Sex was found to influence time to infection, with males having a hazard of testing positive 1.97 times greater than females (95CI: 1.06–3.67, *p* = 0.03) (Appendix A). Age also had an influence, with an increased hazard being found for 12–24-month-olds (Hazard Ratio, HR = 2.12, 95CI: 1.00–4.51, *p* = 0.05), compared to animals under 6 months of age. No interactions were found between explanatory variables.

Considering survival time to death. an interaction was found between treatment set and previous TB history (*p* < 0.0001). In addition to these two variables, and the interaction, age, sex, and being vaccinated as an adult had the lowest wald test values (Table 2) and so were considered for the multivariable model. Due to a violation of the proportional hazards assumption, the multivariable model was split into two time-periods, and the results are displayed separately in Table 3. After this split, proportional hazards assumptions were met for both models (*p* = 0.396 for the earlier model, and *p* = 0.20 for the later time-period).During the first six months, being a member of the HC treatment set increased the hazard of death (HR = 3.34, 95CI: 1.21–9.24, *p* = 0.02), but in the subsequent 19 months, it was found to have a strong protective effect (HR = 0.50, 95CI: 0.29–0.88, *p* = 0.017), when compared to individuals in the control set (Table 3). As the proportional hazards assumption was initially violated in the model of time to death, the time period was split into two blocks (<6 months and 6–25 months) based upon visual assessment of the LOESS line (Appendix A). Following this division, the proportional hazards assumption was met for both models. Of 306 individuals included in the analysis of time to death, 228 (74.5%) were lost during the study period. Previous group history of TB increased the hazard of death in both time periods, whereas the effects of sex and treatment group were found to differ between the two time periods (Table 3). Neither being vaccinated as a pup (HR = 0.82, 95CI: 0.56–1.20, *p* = 0.31), nor as a dominant adult (HR = 0.63, 95CI: 0.32–1.24, *p* = 0.18) affected survival time. An interaction between treatment set and previous history of disease was found, and this interaction term was included in the Cox model.

## 4. Discussion

This study aimed to establish whether there was any empirical evidence for a group-level benefit of trait-based vaccination. We used two different treatment strategies: a high-contact (HC) approach where individuals with the greatest rates of high-risk social interactions were vaccinated, and a high-susceptibility (HS) approach where only the most susceptible members of the population were vaccinated. In both cases, after two years of treatment, there was no significant change in the odds of testing positive for TB in the treatment sets (HC: OR = 1.0, 95CI: 0.33–3.04, *p* = 1.0 and HS: OR = 1.5, 95CI: 0.28–7.91, *p* = 0.63), in contrast to untreated control groups which saw a marked increase in odds (OR = 5.4, 95CI: 0.94–30.98, *p* = 0.058). The increased odds seen in the control groups reflected a general increase in observed clinical cases within the area, so may be reflective of locally increased infection pressures, possibly mitigated against by the vaccination strategies. It was also observed that being a member of a HC treatment group had a protective effect on an individual’s survival times when compared to a meerkat from a control group (Hazard Ratio = 0.5, 95% Confidence Interval: 0.29–0.88, *p* = 0.02). No difference was seen in the odds of testing positive between the two treatment approaches used. It is therefore considered that the large OR increase in the control group is indicative of an increase in infection pressure within the area. No corresponding increase in odds in either of the treatment arms of the study is seen as supporting evidence towards a treatment effect.

Although both treatment strategies appear to have performed well compared to the control, there was little to differentiate the two strategies from each other in terms of benefit. The odds ratios were similar for the two treatments when comparing study start and end, and the hazard ratios for death and first infection were comparable (Table 3, Appendix A). However, far fewer vaccinations were required to achieve these results in the high contact groups. Over the course of the study, 14 dominant individuals were vaccinated (HC), compared to 52 pups (HS). One of the key benefits of trait-based vaccination being explored was the potential to reduce infection transmission with the minimum number of vaccinations administered. By the end of the study period, as a consequence of routine levels of adult mortalities, the majority of the HS treatment groups (81.0%) were made up of animals that had been born within the study period and consequently were vaccinated. Protective effects of treatment may therefore be due to vaccination levels in excess of the HIT, rather than to a success specifically due to a targeted approach. In the HC groups, however, this same effect was achieved with a vaccinated proportion of the group far below the threshold. In terms of efficiency therefore, the HC strategy is a better example of a successful targeted intervention.

In a similar manner, indirect benefits of BCG vaccination against TB have been demonstrated previously in unvaccinated badgers (*Meles meles*), when the social group included vaccinated individuals [48]. The results from the present study of meerkats corroborate the findings in the badger study, but here, common traits were shared by the vaccinated individuals, rather than being randomly distributed. It was originally proposed that we would include a further treatment set in which vaccination was carried out at random within a social group, as with the badgers, however this was not feasible due to a limited meerkat study population. Whilst inclusion of such a treatment set would have allowed us to compare our targeted treatment with a randomised approach, our study question here was to ascertain whether a targeted approach could work at all. Results presented here provide evidence to support our hypothesis that transmission could be successfully reduced by vaccinating individuals likely to be superspreaders, or individuals that are most susceptible to infection.

These treatment strategies were designed to reduce within-group transmission, but the measured risk of new infections (incidence) within each group is affected both by transmission within the group, and introduced infections. Whilst it would have been useful to have information on the origin of each infection, this study was not able to measure the number of new infection incursions into a group. The number of observed clinical cases during the study period was in line with the highest rates previously recorded in the area [26]. This fits with the incidence data from the control groups, and suggests that there were high levels of circulating infection in the study area during this time-period, possibly influenced by environmental factors.

The timing of this work coincided with a drought within the study region. Population numbers dropped dramatically in late 2015, as a result of both adult mortality and a lack of pup recruitment (births), with nearly all reproduction ceasing at the height of the usual breeding season [49]; the increased background hazard of death is observable in Appendix A. Whilst having lost 74.5% of the population during the study appears high, meerkats have been found to have a median life expectancy of 2.3 years [31], and so with a study duration similar to that, this rate is not surprising. These environmental and demographic factors likely explain the differences in treatment effects between the first six months of the study (September 2014–February 2015) and the following 18 months. There was no other obvious biological change after 6 months of the study. Decreased rainfall has previously been shown to be a risk factor for groups showing first signs of TB [26] and the increased number of cases over the 2015–2016 summer supports this association.

Environmental factors may have had an effect upon transmission, but it seems likely that this effect would have been equal for all groups. Social group sizes were matched between the study arms, and immigration levels into groups were similar across the study (immigrations occurred in a single group in each study arm). Furthermore, the structure of all groups was similar, comprising of a dominant pair, with co-operative sub-ordinates of varying ages. For the purposes of inter-group infection transmission, it is likely to be group structure, not size, that drives spread [50] and group size was not shown to be a risk factor for a group becoming infected [26]. It therefore appears that the treatment groups were likely exposed to similar levels of new infection as the controls. Whilst there was a difference observed in the odds of being TB test positive between the control and one of the treatment groups at the outset of the study, the magnitude of this difference increased markedly over the two-year period (Table 1), implying that the risk of becoming infected was truly reduced in the treatment groups compared with controls. Given the high incidence rate of new cases (Figure 1), it seems likely that the increased odds of infection in September 2016 (Table 1) was due to a higher level of within-group transmission in control groups than in the treatment groups, rather than to an increased retention of positive animals within these groups which would have increased the odds, but not incidence rate. We argue that rather than these results representing either differences in the groups’ exposure to internal sources of infection, or differing survival of infected animals, that vaccinations within the treatment groups appear to have succeeded in reducing transmission.

Aside from the group protective effect, perhaps surprisingly, the direct treatment benefit to vaccinated individuals was unclear. Being vaccinated was not shown to have a significant enough effect upon survival time. Unlike in the study of vaccination in badgers [48], vaccinated meerkats were highly reactive to the test of cell-mediated immunity, making interpretation of results from these animals uncertain. As a result, vaccinated animals were excluded from the investigation of time until infection because the diagnostic test results are likely to have been influenced by the vaccination itself. Benefits to vaccinated animals may well have been limited by infection prior to vaccination. Half of the vaccinated dominants had evidence of exposure prior to the intervention. Despite being the most commonly used TB vaccine globally, estimates of BCG’s efficacy vary widely with published figures of between 0% and 80% in humans [51]. The vaccine has greater benefit in slowing progression of disease than for preventing infection [52]. Disease may well have already been progressing in these individuals, thus offering some explanation as to why they did not experience any increased survival times.

The progression from initial infection to active disease is known to take several months in other species [53]. In meerkats this information is not known, but animals as young as 10 months have been recorded with lesions [38], suggesting that progression is possible in less than a year. Even so, if transmission were to be stopped, or reduced, at a fixed time point there would still be a lag phase, likely to be several months, before this manifested as a reduction in clinical disease. This is consistent with the protective effect of the HC treatment being absent during the first six months of the study. Previously infected animals may have continued to develop disease at the same time as uninfected individuals were experiencing a reduction in incidence. This suggests that a study of longer duration is warranted to investigate whether these trends continue.

Given the social nature of meerkats, it is inevitable that social group will have had some influence on an individual’s survival time within the population. Analysis of these survival times was carried out here on an individual animal basis (Table 2 and Table 3), and yet treatment was allocated at a group level. It was therefore not possible to include a random effect variable for social group, as this effect was so closely linked to treatment set. It is possible that the three groups that were randomly allocated to the HC treatment were, by chance, more successful groups in terms of their survival. This problem of allocating treatments to clusters of individuals is recognised in the medical literature, causing a number of issues [54]. Given the similarities between the treatment groups and the control groups in this study, we believe that the results are a reliable indicator of success, but for a more definitive conclusion, a larger study involving more groups is required; such a study appears to be justified given these results.

## 5. Conclusions

In this study, we have found empirical evidence supporting the theory that a trait-based targeted vaccination strategy can be used to reduce transmission of *M. suricattae* within a population of wild meerkats. A delayed onset of the treatment effect and uncertainty caused by randomisation of treatment at the group-level suggest that a further study involving more groups, and over a longer duration, is warranted to substantiate this apparently important finding. Taken alone, we believe that this study provides supporting evidence for the theory that trait-based vaccination could be advantageous in the control of infectious disease and is worthy of further research. The wider implications of this study for disease control are that for any system in which the drivers of transmission are sufficiently understood, a targeted approach using fewer doses may offer a substantial improvement in efficiency when compared to standard protocols. Such improvements in efficiency could make infectious disease control more practically achievable in animal (particularly wildlife) and human populations.

## Figures and Tables

**Figure 1 animals-12-00192-f001:**
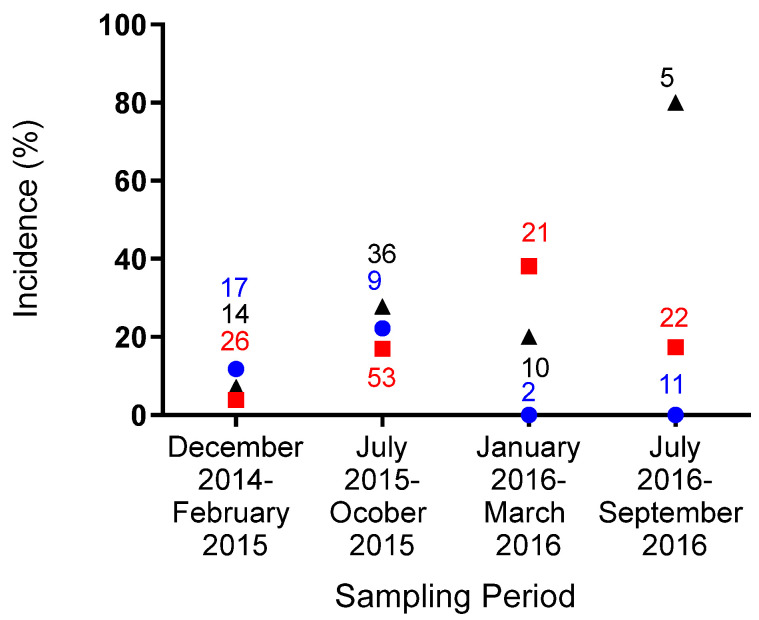
Changes in infection incidence risk over time in groups of wild meerkats receiving targeted or no vaccination. The number of new cases is expressed as a proportion of those eligible to be incident cases in each of the three treatment sets within each three-month block across the study. Numbers on the chart relate to the total number of individuals eligible to become a case within each set, at each time point.

**Table 1 animals-12-00192-t001:** Odds of testing positive for TB in groups of meerkats receiving no vaccination (control) or trait-based targeted vaccination regimes (high susceptibility and high contact). The odds of an individual testing positive (with the sample size in brackets) are shown in the third column for each treatment set at both the start and the end of the study, excluding vaccinated animals. An individual was considered test positive if it generated a positive result on any one of the three diagnostic tests.

Time Block	Treatment Set	Odds of Testing Positive	Odds Ratios		
			Control _Start_	High Susceptibility _Start_	High Contact _Start_
Study start (September–November 2014)	Control	0.83 (22)	-	-	-
	High Susceptibility	0.18 (26)	**4.58 (1.18–17.79, *p* = 0.03)**	-	-
	High Contact	0.45 (29)	1.85 (0.59–5.85, *p* = 0.29)	2.48 (0.66–9.31, *p* = 0.18)	-
Study end (July–September 2016)	Control	4.5 (11)	5.4 (0.94–30.98, *p* = 0.058)	-	-
	High Susceptibility	0.27 (14)	-	1.5 (0.28–7.91, *p* = 0.63)	-
	High Contact	0.45 (29)	-	-	1.0 (0.33–3.04, *p* = 1)

The odds ratios (with a 95% confidence interval and a *p*-value) are shown comparing the treatment sets against each other at the start of the study (in light grey); and the change in each treatment set between the start and end of the study (in dark grey). Significant results are given in bold.

**Table 2 animals-12-00192-t002:** Univariable analysis for survival time to death in 306 wild meerkats.

Variable	Category	HR *	95% Confidence Interval	Wald Test *p*-Value for Variable
Vaccinated as a pup	No			0.305
	Yes	0.82	0.56–1.20	
**Vaccinated as an adult**	**No**			**0.180**
	**Yes**	**0.63**	**0.32–1.24**	
Dominance	No			0.622
	Yes	0.92	0.66–1.28	
**Sex**	**F**			**0.095**
	**M**	**1.26**	**0.96–1.66**	
**Age**	**0–6 months**			**0.020**
	**6–12 months**	**0.55**	**0.37–0.82**	
	**12–24 months**	**0.69**	**0.48–1.00**	
	**>24 months**	**0.87**	**0.61–1.24**	
**Previous history of TB within social group**	**No**			**<0.0001**
	**Yes**	**2.56**	**1.95–3.37**	
**Treatment Set**	**Control**			**0.141**
	**High Susceptibility**	**0.79**	**0.56–1.13**	
	**High Contact**	**1.14**	**0.85–1.53**	

* HR, hazard ratio. Terms retained for multivariable analysis are given in bold (*p* < 0.2).

**Table 3 animals-12-00192-t003:** Multivariable analysis results for survival time to death in 306 wild meerkats. Data are presented for two time periods; the first 6 months of the project, and the subsequent 18 months.

Variable	Category First Period (0–180 Days)				Second Period (181–760 Days)		
	HR *		95% Confidence Interval	*p*-Value	HR *	95% Confidence Interval	*p*-Value
**Sex**	F						
	M	**0.48**	**0.25–0.93**	**0.030**	**1.96**	**1.35–2.84**	**<0.001**
**Age**	0–6 months						
	6–12 months	0.49	0.21–1.13	0.094	0.67	0.39–1.15	0.149
	12–24 months	**0.22**	**0.07–0.76**	**0.016**	0.94	0.59–1.50	0.798
	>24 months	**0.38**	**0.15–0.96**	**0.040**	1.13	0.66–1.94	0.646
**Previous history of TB within social group**	No						
	Yes	**98.75**	**21.35–456.84**	**<0.0001**	**2.14**	**1.18–3.88**	**0.012**
Treatment Set	Control						
	High Susceptibility	2.70	0.99–7.35	0.053	0.72	0.44–1.18	0.188
	High Contact	**3.34**	**1.21–9.24**	**0.020**	**0.50**	**0.29–0.88**	**0.017**

* HR, hazard ratio. Significant findings are highlighted in bold.

## Data Availability

The data used in this analysis are available at https://figshare.com/articles/dataset/Trial_Survival/6142820 (accessed on 5 January 2022).

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
