# Peer review of "Trait-Based Vaccination of Individual Meerkats (Suricata suricatta) against Tuberculosis Provides Evidence to Support Targeted Disease Control"

_animals, 2022, doi:10.3390/ani12020192_

Round 1

Reviewer 1 Report

The paper is well written and its content is particularly significant for wildlife health management.

I have just a few of minor issues to submit to your attention.

Editing/misprints

  • Row 280: “was found” is probably a “leftover” misprint
  • In the caption of table 1, you mention bold characters, as well as dark and light grey, but I cannot see these formats in the body of the table. Maybe it’a a misprint.
  • Readability of table 2.a should be improved. As an example, HR of being vaccinated as an adult is written in bold, but itse p-value is not significant. Same situation for treatment set. This raises a bit of confusion
  • Table 2.b too could be improved by a clearer editing the splitting into the two different study periods
  • For supplementary figure 2, I would suggest to use different degrees of a lighter color than red
  • Supplementary Figure 3 would benefit of an editing improvement, by drawing, if possible, hypothetical zero gradient lines for both the study periods

Concerning the contents, I have not properly issues; rather, a couple of questions/ideas, that may be useful to enrich the discussion (or maybe to raise some doubts – sorry in the latter case):

  • In an hypothetical randomised treatment, how many meerkats would you have randomly treated? The same number scheduled for the targeted one, or more? If more, would the randomised and targeted treatment have been comparable?
  • In the last sentence of the conclusions, You write that the improvement in efficiency provided by targeted vaccination could make disease control more achievable. To a certain extent, namely when the objective is to reduce the impact of a disease, I agree, but I have doubts that this approach could be praticable when the objective is disease eradication

I hope you’ll find my comments useful.

Sincerely

Author Response

The paper is well written and its content is particularly significant for wildlife health management.

I have just a few of minor issues to submit to your attention.

Editing/misprints

Row 280: “was found” is probably a “leftover” misprint

Response: Thank you for reviewing this work and for those positive comments. We can see the error on line 280 and have now amended lines 285-286 to read: “….with no evidence of an increase found in either treatment set (p =0.63 for HS and p=1.0 for HC, Table 1).”.

In the caption of table 1, you mention bold characters, as well as dark and light grey, but I cannot see these formats in the body of the table. Maybe it’a a misprint.

Response: Thank you for pointing this out; you are quite correct, the use of bold, and grey shading had been lost in the edits. A revised version of Table 1 is now included.

Readability of table 2.a should be improved. As an example, HR of being vaccinated as an adult is written in bold, but itse p-value is not significant. Same situation for treatment set. This raises a bit of confusion

Response: Thank you for raising this potential area of confusion. We have edited the Table 2a legend to clarify this situation. Table 2a gives the results of a univariable analysis and 2b gives the subsequent multivariable. The terms in 2a which are in bold are not terms for which the p-value is <0.05, but terms where the value is <0.2 and thus the terms carried forward to the multivariable. The legend on lines 362-365 now reads “An interaction was found between treatment set and previous TB history (p <0.0001). These two variables, and the interaction, were included in the final multivariable model, along with age, sex, , and being vaccinated as an adult. These terms are given in bold.”.

Table 2.b too could be improved by a clearer editing the splitting into the two different study periods

Response: We have inserted a separation between the time periods in table 2b as you have suggested and agree that this does look clearer.

For supplementary figure 2, I would suggest to use different degrees of a lighter color than red

Response: We agree with the reviewer that supplementary figure 2 is rather heavy in appearance and have changed the extinct category from black to white in the revised manuscript. We believe that this lightens the whole figure.

Supplementary Figure 3 would benefit of an editing improvement, by drawing, if possible, hypothetical zero gradient lines for both the study periods

Response: A hypothetical line has been added to Supp. Fig 3 and the following information added to the legend in lines 393-396: “A hypothetical zero-gradient line is illustrated as a broken red line, demonstrating that it was not possible to draw this within the 95% confidence interval of the LOESS line. This is evidence of non-proportional hazards and it was therefore necessary to split the model into two time periods.”.

In an hypothetical randomised treatment, how many meerkats would you have randomly treated? The same number scheduled for the targeted one, or more? If more, would the randomised and targeted treatment have been comparable?

Response: This is a very interesting question, thank you! A randomised treatment set of groups would have been very interesting to include and we had originally intended to do so. Had we had access to more social groups we would certainly have done so (see lines 438-441). In a hypothetical treatment group we would have vaccinated at the level of the HIT (Herd Immunity Threshold) to simulate a real world ideal. The HIT (HIT = 1- 1/R0) is unknown for this infection in meerkats, but based on estimations of 1.1 in cattle and 1.16 in badgers (Cox et al. 2005, Smith, 2005). Ideally our own HIT would have been based on a modelling exercise, and would partially have depended on exactly what question we were trying to answer by including it. It is likely, based on those badger and cattle estimate, that the vaccination coverage that we were aiming at would be relatively low. We think that it is important to clarify that in this particular study, the advantage of the targeted treatment which was are looking to exploit is not that it works better at controlling disease than the randomised method, but that it is more achievable. Hence in lines 400-401 and 424-426 we are careful to highlight that the aim of the study was to see if there was any group benefit at all of the targeted approach, not to compare it against a randomised treatment.

Cox, D. R., Donnelly, C. A., Bourne, F. J., Gettinby, G., McInerney, J. P., Morrison, W. I. & Woodroffe, R. 2005. Simple model for tuberculosis in cattle and badgers. Proceedings of the National Academy of Sciences of the United States of America, 102, 17588-17593.

Smith, G. C. 2005. Modelling bovine tuberculosis in wildlife and cattle. In: SMITHE, L. T. (ed.) Focus on tuberculosis research. New York: Nova Science Publishers Inc.

In the last sentence of the conclusions, You write that the improvement in efficiency provided by targeted vaccination could make disease control more achievable. To a certain extent, namely when the objective is to reduce the impact of a disease, I agree, but I have doubts that this approach could be praticable when the objective is disease eradication

Response: We agree. We have stated throughout this work that the objective is to reduce transmission (eg lines 424-426), but were the aim total eradication, a more heavy-handed approach would be necessary.

Thank you for all of the comments that you have made, and for the time taken to review this work.

Reviewer 2 Report

This paper provides the scientific basis for the control wildlife diseases in a systematic manner and can be duplicated in other populations as well.

There are some comments to improve the draft of this manuscript.

Author Response

This paper provides the scientific basis for the control wildlife diseases in a systematic manner and can be duplicated in other populations as well.

There are some comments to improve the draft of this manuscript.

Response: Thank you for taking the time to read our submission and for these encouraging comments. We have addressed all of your points, as described below, and believe that these clarifications will improve the manuscript.

Page 3: suggestion = The purpose of this study was to determine whether there was any empirical evidence to demonstrate that  a trait-based system could be effective in a wild Kalahari meerkat population.

Response: Thank you for this suggestion. Your suggested sentence summarises the study and standing alone is clearer than our original suggestion. However, we have discussed this study with a number of colleagues and are frequently met with the assumption that we were trying to draw a comparison between the trait-based approach and a randomised intervention. Given that level of misunderstanding, we felt it important to address not only what our aim was, but to clarify what we were not trying to achieve!

Page 3: Insert a map to show the study area to benefit the global researchers

Response: There is a wealth of background information that we could provide to this study site and we have given the Clutton-Brock and Manser (2016) reference (as well as the site co-ordinates) in order to provide as much of that information. We did not feel that further specifics were directly relevant to this particular manuscript.

Clutton-Brock TH, Manser, MB (2016) Meerkats: Cooperative breeding in the Kalahari. In Koenig WD, Dickinson JL. (Eds) 621 Cooperative breeding in vertebrates: Studies of ecology, evolution, and behaviour (pp.294-317). Cambridge, UK: Cambridge 622 University Press.

Page 3: Did you mark these animals?

Response: We had not previously stated this, so thank you for highlighting that omission. All individuals were already identifiable by applied marking, and as such, we have edited lined 137-139 to read: “All animals included in the study were part of the Kalahari Meerkat Project’s (KMP) long-term study of a free-ranging meerkat population, and as such all were individually identifiable from pre-existing dye marks”.

Page 4: 0.5ml

Response: Thank you, we have inserted this space (line 162).

Page 5: What sort of controls implemented? Describe the environmental conditions during the study

Response: We have edited lines 194-195 to highlight that the control set was made up on unvaccinated groups: “No vaccinations were carried out in the control groups, but this treatment set differed in no other way from either the HC or HS sets”. All three treatment sets experienced drought during this time and that is described in the paragraph beginning on line 456.

Page 6: There was only one permit shown as attachment, you mention of two permits in here.

Response: We had a single ethical permission for this study, issued by the University of Pretoria, and that has been submitted as an attachment. We would be happy to submit confirmation of the other permissions if they are required in addition to the ethics permit.

Page 8: 2015?

Response: Thank you. We have clarified this, and in lines 309-311, now state: “…there were more clinical cases of TB in the meerkat population during the second year of the study (Sep 2015- Aug 2016) than the first (Sep 2014-Aug 2015)…”.

Page 10: Full list of individuals can be presented as supplementary information

Response: Details on all individuals in the study are given in the spreadsheets available at https://figshare.com/articles/dataset/Trial_Survival/6142820. This link is given in the data accessibility section on lines 562-563.

Page 13: how many deaths?

Response: These social groups are dynamic with regular births, deaths, immigration, and emigration. A single figure giving the number of deaths would not give a good indication of the change in the groups over time. We think that Supplementary Figure 2 is much more useful for addressing this question.

Page 13: Unpredictable factor during the onset of the study which may add bias on the results?

Response: We agree that the environmental factors may well have influenced the results, but to the best of our knowledge, all social groups in all three treatment sets would have been exposed to the same conditions. We feel that it was important that this study was a “real-world” example to demonstrate the applicability of findings.

Page 13: what gaps for further research?

Response: We have suggested, in lines 521-524, that a larger study would be worth pursuing, and that the results presented here are make that a justifiable project.